# DECOUPLED REPRESENTATION AND POLICY ACQUISITION FOR CONTINUAL REINFORCEMENT LEARNING

## ABSTRACT

This contribution proposes adiabatic reinforcement learning (ARL), a new method for continual reinforcement learning (CRL). In CRL, we assume a non-stationary environment partitioned into *tasks*. To avoid catastrophic forgetting (CF), RL requires the use of large replay buffers, which leads to very slow learning and high memory requirements. To remedy this, we propose adiabatic reinforcement learning (ARL), a wake-sleep method that performs slow learning of internal representations from high-error transitions during sleep phases. Wake phases are used for the fast learning of policies, i.e., mappings from representations to actions, and to collect new high-error transitions. Representation learning is performed by *adiabatic replay* (AR), a recent CL technique we adapted to the RL setting. AR uses selective, internal replay of samples that are likely to be affected by forgetting. Since this process is conditioned on incoming samples only, its has constant time-complexity w.r.t. tasks. Other benefits include fast adaptation to new tasks, and a very low memory footprint due to the complete absence of replay buffers.

## 1 INTRODUCTION

This article is in the context of continual reinforcement learning (CRL), a branch of reinforcement learning (RL) where a non-stationary environment is assumed in addition to non-stationary observations due to ongoing exploration and model adaptation. Non-stationary data distributions cause the well-known catastrophic forgetting (CF) effect McCloskey & Cohen (1989) when employing DNN learners. The study of machine learning from non-stationary data distribution is the objective of *continual learning* (CL), a particular goal being the mitigation or avoidance of catastrophic forgetting. Both in CL and in CRL, a common simplification is to assume the existence of *tasks*, i.e., phases of stationary data distribution or environment, with non-stationarities (or shifts) occurring only at task onsets, see fig. 1 for a visualization.

### 1.1 MOTIVATION

Since CF is an issue in RL even when environments are stationary, the typical approach is to use large *replay buffers* for storing past samples and replaying them for to the learner. This simulates a stationary distribution, but comes at a significant memory overhead and slows down learning of new

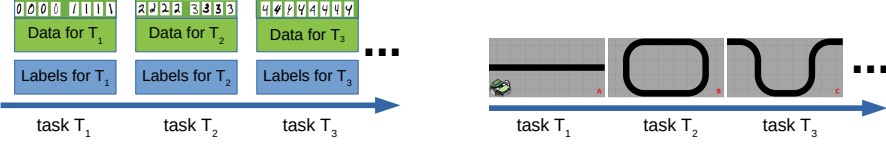

Figure 1: Exemplary CL (left) and CRL problems (right) subdivided into tasks of stationary data distribution or environment. CL problems are usually taken to be supervised classification problems, and non-stationarities can be modeled as shifts in the occurring sample classes (class-incremental learning scenario, see Van de Ven et al. (2022); Bagus et al. (2022)), shown here for MNIST. CRL problems usually define tasks by shifts in environment properties or reward structure. Here, a robotic agent needs to follow a black line, the shape of which changes for each new task.

tasks in CRL since new-task samples need time to sufficiently populate the buffer before they can have an impact. Techniques like prioritized experience replay (PER, Schaul et al. (2015)) attempt to accelerate convergence by focusing on high-error samples, but this requires extensive parameter tuning and can again lead to forgetting if the original data distribution is changed too much Pan et al. (2022).

A CL technique to replace replay buffers in CRL is generative replay (GR), for which new samples are immediately used for training, and which avoids forgetting by mixing them with generated samples from past tasks. This improves learning speed for new tasks, but the replay of samples conditioned on past tasks leads to an ever-increasing number of replayed samples as the number of tasks grows (see Krawczyk & Gepperth (2024b) for a discussion of why such balanced replay is required). Furthermore, this strategy requires that task onsets be known with a high degree of certainty, which is a highly artificial assumption, just as the assumption of distinct, crisply defined RL tasks is in general.

## 1.2 CONTRIBUTION

The contribution of this article is the method of adiabatic reinforcement learning (ARL), which is validated on task-based CRL benchmarks (see fig. 1) without being informed about task onsets at any point. The core of ARL is a replay-based CL method termed adiabatic replay (AR, see Krawczyk & Gepperth (2024a)), which performs generative replay not conditioned on a task, but on incoming samples themselves (*selective replay*) and therefore does not depend on the number of previously encountered tasks. The internal representation of the learner employed here is a fully probabilistic Gaussian Mixture Model (GMM) that supports *selective updating* exclusively with high-error samples. This decomposition of the learner into representation and (policy) readout blocks is typical of state representation learning, see section 1.3. A wake-sleep training algorithm is applied so that no interference can occur between these blocks.

In particular, this article proposes the following contributions to the field of CRL:

- fast learning of new tasks
- extremely small replay buffers and low memory footprint
- generative replay approach focusing on high-error transitions without forgetting
- constant time complexity w.r.t. the number of CRL tasks
- plastic representation over time, arbitrary number of tasks

## 1.3 RELATED WORK

When data distributions are not stationary or samples are not i.i.d. Lesort et al. (2021) catastrophic forgetting (CF, McCloskey & Cohen (1989); Ratcliff (1990)) will occur. Especially deep neural networks (DNNs) are highly susceptible to that rapid performance degeneration Pfülb & Gepperth (2019). Various approaches have been introduced for addressing CF, see Shaheen et al. (2021); Qu et al. (2021); Wang et al. (2023a;b) for extensive surveys. The "default" CL scenario for supervised CL Bagus & Gepperth (2022), also known as *class-incremental learning* (CIL, van de Ven & Tolias (2019); Masana et al. (2022); Zhou et al. (2023)) is adopted in the majority of recent publications. Essentially, it is posited that non-stationary data streams can be partitioned into a subset of non-contradictory, non-overlapping *tasks* with stationary statistics, whose onset and duration are known. However, not all CL methods are suitable for RL, as it differs considerably from the common CIL setting Lesort et al. (2020); Khetarpal et al. (2020); Bagus & Gepperth (2022). We will therefore focus on rehearsal or experience replay (ER, see Rolnick et al. (2019)) and pseudo-rehearsal or generative replay (GR, see Shin et al. (2017); Kamra et al. (2017); Atkinson et al. (2018a)), which either store samples from previous tasks or use a generator to obtain them in unlimited quantities. Such samples are then merged with current samples to avoid forgetting. ER and GR in particular have become strong baselines Balaji et al. (2020); Zhang et al. (2022) in CL and work in a variety of scenarios Verwimp et al. (2021); Hayes et al. (2021).

Continual reinforcement learning (CRL) is studied for a variety of algorithms, of which the most commonly used is deep Q-learning (DQN) with experience replay, see, e.g., Mnih et al. (2013). However, more sophisticated variants like soft actor-critic (SAC) are studied as well Wolczyk et al.

(2022b). An overview concerning the field of CRL is given in, e.g., Lesort et al. (2020); Khetarpal et al. (2020); Shaheen et al. (2021). There are several works on generalizing CL methods to CRL: a straightforward adaptation of pseudo-rehearsal is described in Atkinson et al. (2018b), where standard double DQN with experience replay is employed for training an short-term memory (STM) instance on the current task. The trained STM instance, together with the (large) replay buffer from the current task, are then used to train a long-term memory (LTM) instance about the current and past tasks. This is convincingly demonstrated for a sequence of three Atari games. A drawback of this work is the need to know about task onsets, and of course that this will scale linearly with the number of tasks, at least for a large number of tasks. A similar approach is adopted in the S-Trigger model Caselles-Dupré et al. (2019), which, in addition, detects task onsets by using the VAE as an outlier detector and thus no longer needs to be informed about task onsets. Linear scaling behavior w.r.t. time will still be an issue, and new tasks may not be associated with a change in observations but in the optimal policy, thus limiting applicability to a subset of possible CRL problems. S-trigger belongs to the class of *state representation learning* models (see Lesort et al. (2018) for an overview), which decompose RL into learning a representation for observations and for policies. Similar in this respect is DARLA Higgins et al. (2017), which uses VAEs for this purpose as well and shows that such an approach can provide TL with invariance to unforeseen variations. DARLA is not intended for CRL, although it is unclear how invariance can be controlled and restricted. Finally, the DisCoRL model Traoré et al. (2019) employs a babbling phase (purely random exploration) for learning a model encoding representational states with some degree of invariance. This is a very significant contribution, albeit limited to cases where random exploration will reach all possible observational states, which is unlikely to be the case in general settings.

Little consensus exists concerning the benchmarks on which to evaluate CRL. Commonly used benchmarks are Atari gamesAtkinson et al. (2018b), Continual World Wołczyk et al. (2021), but mostly self-defined benchmarks Traoré et al. (2019); Daniels et al. (2022); Tomilin et al. (2024).

## 2 METHODS

We implement three benchmarks using the Gazebo (Harmonic) simulator [1] and the gz-transport package for controlling it from Python. All learning algorithms are self-implemented in Python3 using TensorFlow 2.14. The source code for the experiments is publicly available[2].

### 2.1 BENCHMARKS

The simulated robot is modeled after the popular $3\pi$ robot from Pololu Robotics, see fig. 3. It is controlled by a differential drive, with two wheels (radius: $\approx 1.55\,cm$, separation: $\approx 9\,cm$) driven by independent motors. In addition, there is a passive caster wheel for balancing. Inputs to the differential drive are wheel speeds $v_L$ and $v_R$ measured in meters per second. Observations and commands are exchanged at a fixed frequency of 15Hz (in simulation time). An RGB camera with an aperture of $50\,\deg$ can be placed at the front of the robot. The action space consists of discrete actions $a_t \in \mathbb{N}_0^+$, each action being defined by a 2-tuple of wheel speeds $(v_L, v_R)$.All friction parameters and drive speed/acceleration limits are set such that wheel speed commands are realized quasi-instantaneously without wheel slips or gliding.

### 2.1.1 LINE-FOLLOWING (LF)

This benchmark is represented by four different racetracks consisting of a black circle drawn on differently colored ground planes, see fig. 2. The goal for the robot (placed on the black line) is to keep the left border of the black line as centered as possible in the camera image while moving forward, and thus, to follow the line. We define four successive CRL tasks between which environmental shifts occur: LF1(red ground plane), LF2 (green ground plane), LF3 (blue ground plane) and LF4 (yellow ground plane). An environment shift corresponds to placing the robot on a designated spawn point on a different racetrack. Observations $\vec{o}_t$ are formed by slicing the middle 4 rows of each received $100 \times 100$ image and concatenating the 3 last $100 \times 4$ images along the row axis. The first observation is duplicated during concatenation: twice for the first iteration, and once for

---

[1] www.gazebosim.org

[2] https://github.com/anon-scientist/iclr25-arl

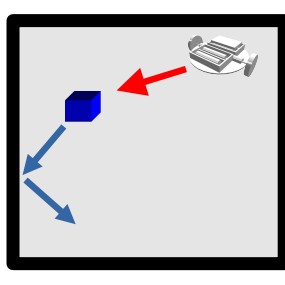
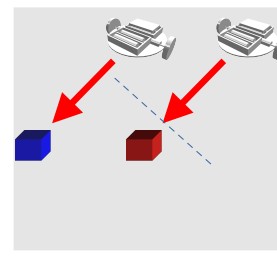

Figure 2: Benchmarks used in this study. Left: line-following(LF, only first task shown) where the robot must follow a black line on differently colored ground planes. Middle: robot pursuit(RP, only task 1 shown), where the robot must follow a moving block of varying color without transgressing the boundaries of the arena indicated by black lines. Right: pushing-objects(PO, tasks 1 and 2 are shown) where the robot must approach an inert block of varying color, and either push it, or stop in front of it.

| ↓Benchmark - action→ | 0 | | 1 | | 2 | | 3 | | 4 | |
|---|---|---|---|---|---|---|---|---|---|---|
| Line-Following | 0.05 | 0.35 | 0.15 | 0.25 | 0.2 | 0.2 | 0.25 | 0.15 | 0.35 | 0.05 |
| Pushing-Objects | 0 | 0 | 2 | 0.4 | 0.4 | 2 | 1.2 | 1.2 | - | - |
| Robotic Pursuit | 0 | 0 | 2 | 0.4 | 0.4 | 2 | 1.2 | 1.2 | - | - |

Figure 3: Left: the simulated two-wheeled robot which can be equipped with a front camera. Right: all benchmarks assume that the robot can perform 4 discrete actions, defined by the given left/right wheel speed pairs per action.

the second iteration, so there is always a valid observation available. A terminal state occurs when image processing is unable to detect the left edge of the line in the image, or when the maximum sequence length of 25000 is reached. The actions space comprises 5 distinct actions: $a_t \in [0, 4]$ corresponding to four different speeds for left and right, as well as a single action for pure forward acceleration, see fig. 3. The dense reward signal $r(t)$ is calculated from the deviation $d(t)$ (in pixels) of the left edge of the line from the center of the image $\vec{o}_t$ of width $W = 100$ and is normalized between $[0.0, 1.0]$: $r_t = 1 - \left| \frac{d - \frac{W}{2}}{\frac{W}{2}} \right|$, with $d \in [0, W]$. However, terminal states (left edge of the line no longer in image) are penalized by a value of $-1.0$. The reward function does not reward or penalize different speeds.

### 2.1.2 PUSHING-OBJECTS (PO)

In this benchmark, the robot is placed in front of one out of several colored cubes (see fig. 2) and ends with the robot losing visual contact, pushing the cube or reaching the maximum number of 30 actions. Pushing or stopping should depend on a cube's mass (tied to color): massive cubes must not be touched/pushed, whereas massless cubes should be pushed. The robot is initially tilted $\pm 15$ degrees away from the cube it is facing, so a purely random walk will not bring it, on average, near the cube. This benchmark consists of four tasks PO1 - PO4 separated by an environment shift that modifies cube colors (red, blue, green, then yellow) and masses (20kg,0,0, then 20kg). Observations $\vec{o}_t$ are $100 \times 100$ RGB images downsampled to $20 \times 20$ size obtained by the robot's forward-looking camera. The reward $r_t = A(\vec{o}_t, a_t) + B(t)$ is composed of two terms, of which $A(\vec{o}_t, a_t)$ is given continuously, and $B(t)$ only for a terminal state. Terminal states are reached either after 30 iterations, when the robot loses sight of the cube, or when the robot touches one. Approach behavior is encouraged in all tasks by $A(\vec{o}_t, a_t) = 1 - |\mu_x - 10|$, where $\mu_x$ is the $x$ component of the center-of-gravity of non-background cube pixels in $\vec{o}_t$. $B(t)$ depends on a cube's mass: $B(t) = 10$ when touching a massless cube, and $B(t) = -10$ when touching a massive one, and $B(t) = 0$ when no cube is touched. When the robot loses sight of the object (only background pixels in the image), a small punishment is given: $B(t) = -1$. The robot's action space consists of four actions: forward, stop, left and right, each with different speeds as shown in fig. 3, resulting in a total of 4 discrete actions denoted as $a_t \in [0, 3]$.

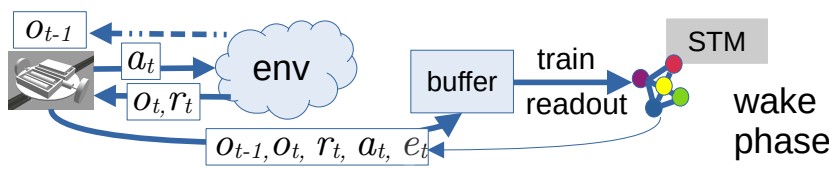

Figure 4: ARL during a wake phase. The readout layer of the learner $\mathcal{S}$ is updated with transitions from the buffer, while the generator is frozen. At the same time, $\mathcal{S}$ is used to compute the TD error of transitions that are stored in the buffer.

### 2.1.3 ROBOTIC PURSUIT(RP)

In this benchmark, the robot is supposed to follow a moving object whose color and shape varies across tasks, see fig. 2, without transgressing the borders of the arena. The robots goal is to reach and catch the moving object by touching it regardless of shape and color. The benchmark ends if the robot looses track of the object, touches the object, reaches the maximum number of 80 actions or leaves the arena. The robot is placed at one side of the arena tilted away from the center by $\pm 15$ degrees, while the moving object is placed in the center. This benchmark consists of four tasks RP1 - RP4 that differ in their objects to follow (red cube, green capsule, blue sphere then yellow cylinder). Observations $\vec{o}_t$ are $100 \times 100$ RGB images downsampled to $20 \times 20$ size obtained by the robot's forward-looking camera. The reward structure is the same as for the PO benchmark, with the exact same structure for $A(\vec{o}_t, a_t)$, but with a slightly modified $B(t)$. When touching the robot the reward is always $B(t) = 10$. There is an additional terminal state of leaving the arena which results in a reward $B(t) = -1$. This terminal state is triggered when the robot is touching the bounding box of the arena. A reward of $B(t) = -10$ is given when the robot reaches the maximum number of actions without touching the object. The robot's action space is the same as in the PO benchmark as shown in fig. 3. The moving object in the scene only ever moves forwards at a constant speed that is slightly slower then the robot and gets redirected with a random angle when touching the bounding box of the arena.

### 2.2 BASELINES

One set of evaluation baselines relies on vanilla deep Q-learning (DQN) and double deep Q-learning (DDQN) with experience replay (ER) using a buffer of size $M$. We test several values for $M$ for each benchmark such that the buffer is either much larger than one task's worth of samples, or much smaller. If the buffer is large, then we should expect that it can mitigate CF, and the reverse should be the case for a small buffer. Conversely, small buffer sizes should enable fast learning. Sampling from the buffer is performed either uniformly Vitter (1985) or via prioritized experience replay (PER) Schaul et al. (2015). All DQN methods are realized by a three-hidden-layer DNN with 256 units in each layer. Exploration is performed in an $\epsilon$-greedy fashion, where $\epsilon$ is decreased from an initial value of $\epsilon_0$ by the equation $\epsilon_t = \epsilon_{t-1} - \Delta_\epsilon$. We tune $\Delta_\epsilon$ such that $\epsilon = 0.2$ at the end of each task, and set $\epsilon_0$ to 1.0 before the first task, and to 0.5 at the start of tasks $n > 1$ in order to re-use existing knowledge where possible and feasible. For prioritized experience replay, we use consensus parameter values $\alpha = 0.6$ and $\beta = 0.6$ and perform linear annealing of $\beta$ such that its value reaches 1 at the end of each task. All DNNs are trained using the Adam optimizer with a learning rate of 0.001 and a mini-batch size of 32. Update frequencies for DDQN are always 200 iterations. The discount factor for Q-learning is always set to $\gamma = 0.8$.

For completeness, we also investigate sequential fine-tuning (SFT) to adapt to new tasks, using no replay buffer but rather a DNN learning rate reduced by a factor of 10.

### 2.3 ADIABATIC REINFORCEMENT LEARNING (ARL)

**ARL foundations** ARL relies upon the adiabatic replay technique described in Krawczyk & Gepperth (2024a). An AR instance is composed of a GMM layer computing $p(\boldsymbol{x}; \{\boldsymbol{\theta}_k\}) = \sum_{k=1}^{K} \pi_k \mathcal{N}(\boldsymbol{x}; \boldsymbol{\theta}_k)$ termed *generator* connected to an affine readout layer $r(\boldsymbol{x}; W, \boldsymbol{b}) = W\boldsymbol{x} + \boldsymbol{b}$ termed *solver*. The solver operates on the vector of posterior probabilities computed by the genera-

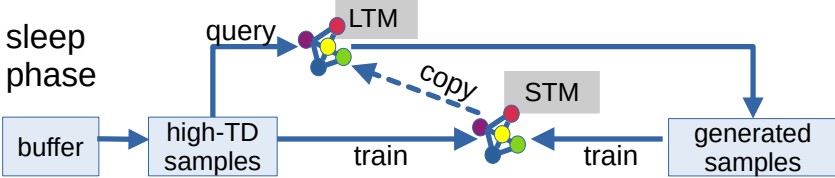

Figure 5: ARL during a sleep phase. Initially, the learner $\mathcal{S}$ (see text) is copied to a frozen long-term memory instance. The long-term memory selectively replays observations and pseudo-targets during sleep phase learning, while the short-term memory (both generator and readout layer) is updated on high-TD-error samples collected in previous wake phases.

tor for the current sample as $\gamma_j(\boldsymbol{x}) = \frac{\mathcal{N}(\boldsymbol{x};\theta_j)}{\sum_k \mathcal{N}(\boldsymbol{x};\theta_k)}$. The generator optimizes a log-likelihood-loss by SGD from random initial conditions as described in Gepperth & Pfülb (2021), whereas the solver independently optimizes an MSE loss. Main AR parameters are the number of generator components $K$, as well as the mini-batch size $\beta$ for SGD training. Importantly, training GMMs by SGD requires to define an "initial adaptation radius" $\sigma_0$. This quantity controls how many adjacent GMM components are adapted during a gradient descent step (none for $\sigma_0 = 0$) and is reduced to 0 during SGD training based on an automatic control scheme. For enabling CL, an AR instance performs *selective sampling*, inputting a query $\boldsymbol{q}$ and producing a sample $\hat{\boldsymbol{q}}$ that the generator considers similar to the query in the sense that they are both, with high probability, generated from the same component: $\mathrm{argmax}_k \gamma_k(\boldsymbol{q}) = \mathrm{argmax}_k \gamma_k(\hat{\boldsymbol{q}})$. Both generator and solver are then trained on generated and new samples, which restricts forgetting.. The goal behind selective sampling is to limit replay to samples from past tasks that are likely to be overwritten by new data, and thus require protection.

**ARL architecture** An ARL experiment is subdivided into tasks of length $T$, at the onset of which environment statistics change. Generally, an ARL agent is not informed about this. ARL learning is conducted in alternating wake-sleep phases, which together form a learning *cycle*. An ARL agent consists of an AR learner $\mathcal{S}$ which models the transitions of the current cycle $c$, $\mathcal{D}_c$, while retaining previous knowledge $\mathcal{D}_{1:c-1}$ due to selective sampling. In wake phases of length $C < T$, the agent explores its environment according to the chosen exploration strategy and stores transitions $(\vec{o}_t, \vec{o}_{-1}, a_t, r_t)$ in a buffer together with their TD error $e_t$ measured by $\mathcal{S}$, which is measured by the learner $\mathcal{S}$. This is visualized in fig. 4. At the same time, the solver (not the generator) of the AR learner $\mathcal{S}$ is updated with transitions from the buffer irrespectively of TD error. Before sleep phases, the initial adaptation radius $\sigma_0$ is set to a pre-determined value, a percentage $\chi$ of highest-error transitions is selected from the buffer and the learner is copied to a long-term memory $\mathcal{L}$ such that the long-term memory represents past-cycle data: $\mathcal{L} \sim \mathcal{D}_{1:c-1}$. Then, the generator of $\mathcal{S}$ is trained with high-error transitions from the buffer until convergence, i.e., when the adaptation radius $\sigma(t)$ (see Gepperth & Pfülb (2021) for details) reaches a predetermined value. Subsequently, the buffer is cleared. This is visualized in fig. 5.

**Reasoning behind ARL design choices** A common assumption in CL literature, see section 1, is that data $\mathcal{D}_n$ for the current task $n$ is immediately available , meaning that selective sampling can be performed before adaptation takes place. For ARL, cycles take the place of tasks, and data acquisition is sequential. If generator adaptation were performed in parallel to selective sampling, at some point generated samples would no longer reflect the statistics of past cycles, but of the current cycle $c$ as well. Selective sampling using the generator of the frozen $\mathcal{L}$ instead of the plastic $\mathcal{S}$ ensures sampling from the right distribution: $\boldsymbol{q} \sim \mathcal{D}_{1:c-1}$. If $\mathcal{L}$ is to generate samples that $\mathcal{S}$ will likely confuse with incoming ones, these two instances should not diverge too much during a sleep phase. Therefore, wake phases should not collect too many examples and therefore be rather short: $C << T$.

## 2.4 EVALUATION MEASURES

We use the performance measures for CRL defined in Denker et al. (2024), most notably the final accuracy measure $P$ and the forgetting measure $F$. For some experiments, we also tabulate the performance measure $P_{mn}$ where task $m$ is evaluated after having completed training on task $n$.

## 2.5 EXPERIMENTS

Experiments are conducted on a cluster of 40 machines equipped with nVidia GTX3080 GPUs. One experiment takes approximately 2 hours. All results are averaged over three identical runs. Tasks in all benchmarks have a duration of $T = 5000$ iterations.

## 2.6 MAIN CRL EXPERIMENTS

In this set of experiments, we compare the performance of ARL to the baselines described in section 2.2, using the three benchmarks outlined in section 2.1. As buffer size for the baselines, we choose 1000, 5000, 15000 and 50000. Together with the choice of default or prioritized experience replay, this gives us 8 baselines which we denote DDQN+ER/PER-$M$ where $M$ denotes the buffer. The PER parameters were found using grid-search for $\alpha,\beta$ and the annealing rate for $\beta$. We always use double DQN for the baseline experiments. In addition, we test against sequential fine-tuning (SFT) denoted as DDQN-SFT. ARL experiments contain a motor babbling phase of 10000 iterations to initialize the generator, and use the best-practice parameters from Gepperth & Pfülb (2021), or else the following parameters: $K = 324$, $\sigma_0 = 1.$, $C = 2500$, $\chi = 0.1$. The ratio between generated and observed transitions is 2. Evaluation measures are the ones described in section 2.4.

| baseline | task $n$ | | | | | | | | | | | | | | | | |
|---|---|---|---|---|---|---|---|---|---|---|---|---|---|---|---|---|---|
| | | 1 | 2 | 3 | 4 | 1 | 2 | 3 | 4 | 1 | 2 | 3 | 4 | 1 | 2 | 3 | 4 |
| | | | | | | | | | Pushing-Objects (PO) | | | | | | | | |
| DDQN+ER | 1 | 6.94 | 7.71 | 7.7 | 0.62 | 7.72 | 9.99 | 8.4 | 0.66 | 7.91 | 7.73 | 7.72 | 4.31 | 8.33 | 8.25 | 9.81 | 8.73 |
| | 2 | - | 18.39 | 18.51 | 11.46 | - | 18.71 | 18.07 | 18.21 | - | 18.38 | 18.34 | 18.3 | - | 18.19 | 18.61 | 18.85 |
| | 3 | - | - | 7.56 | 1.22 | - | - | 5.93 | 6.34 | - | - | 8.45 | 8.16 | - | - | 9.13 | 9.62 |
| | 4 | - | - | - | 18.44 | - | - | - | 18.16 | - | - | - | 7.94 | - | - | - | 9.27 |
| DDQN+PER | 1 | 7.26 | 9.61 | 9.55 | -1.28 | 7.39 | 8.9 | 7.35 | 1.37 | 7.59 | 8.73 | 8.73 | 7.44 | 8.02 | 8.09 | 7.89 | 9.47 |
| | 2 | - | 16.69 | 15.23 | 13.87 | - | 18.49 | 18.56 | 14.77 | - | 16.68 | 18.45 | 18.56 | - | 14.88 | 18.37 | 18.85 |
| | 3 | - | - | 7.35 | 2.63 | - | - | 7.35 | 9.6 | - | - | 9.33 | 2.14 | - | - | 6.68 | 6.33 |
| | 4 | - | - | - | 12.22 | - | - | - | 14.54 | - | - | - | 13.66 | - | - | - | 8.28 |
| | | | | | | | | | Line-Following (LF) | | | | | | | | |
| DDQN+ER | 1 | 19.45 | 12.53 | 5.53 | 8.3 | 23.68 | 22.07 | 9.58 | 14.14 | 22.02 | 22.32 | 22.11 | 22.17 | 21.71 | 20.40 | 18.78 | 22.67 |
| | 2 | - | 21.01 | 12.87 | 11.83 | - | 20.22 | 16.88 | 14.24 | - | 20.51 | 20.49 | 22.08 | - | 18.16 | 18.3 | 22.3 |
| | 3 | - | - | 22.31 | 20.72 | - | - | 22.47 | 21.16 | - | - | 14.62 | 20.55 | - | - | 17.9 | 19.4 |
| | 4 | - | - | - | 23.58 | - | - | - | 20.93 | - | - | - | 21.2 | - | - | - | 18.69 |
| DDQN+PER | 1 | 22.42 | 17.13 | 8.1 | 13.35 | 20.58 | 23.47 | 12.68 | 13.59 | 21.12 | 21.17 | 22.96 | 17.23 | 23.71 | 22.3 | 21.89 | 22.64 |
| | 2 | - | 17.99 | 19.26 | 6.42 | - | 20.15 | 19.09 | 19.54 | - | 18.01 | 21.58 | 19.93 | - | 18.0 | 21.58 | 19.93 |
| | 3 | - | - | 21.96 | 14.45 | - | - | 18.07 | 21.13 | - | - | 21.05 | 21.97 | - | - | 21.05 | 21.97 |
| | 4 | - | - | - | 21.61 | - | - | - | 22.83 | - | - | - | 21.64 | - | - | - | 21.64 |
| | | | | | | | | | Robot Pursuit (RP) | | | | | | | | |
| DDQN+ER | 1 | 39.26 | 58.12 | 32.76 | 43.31 | 39.72 | 42.74 | 55.49 | 43.65 | 38.59 | 42.54 | 33.79 | 45.48 | 49.96 | 50.77 | 41.51 | 44.74 |
| | 2 | - | 58.67 | 44.88 | 37.1 | - | 47.96 | 55.91 | 54.48 | - | 50.68 | 34.79 | 31.43 | - | 39.98 | 43.23 | 55.26 |
| | 3 | - | - | 47.43 | 31.62 | - | - | 51.5 | 48.66 | - | - | 33.04 | 31.46 | - | - | 45.13 | 54.24 |
| | 4 | - | - | - | 36.62 | - | - | - | 44.72 | - | - | - | 40.18 | - | - | - | 44.07 |
| DDQN+PER | 1 | 48.42 | 43.59 | 59.08 | 37.54 | 49.12 | 44.15 | 32.08 | 43.85 | 38.56 | 50.88 | 42.57 | 51.21 | 37.49 | 44.74 | 52.56 | 53.95 |
| | 2 | - | 53.17 | 44.87 | 31.11 | - | 40.62 | 40.29 | 56.69 | - | 46.73 | 36.52 | 58.66 | - | 39.58 | 56.17 | 57.3 |
| | 3 | - | - | 42.36 | 28.2 | - | - | 41.26 | 49.61 | - | - | 24.35 | 49.33 | - | - | 48.63 | 52.9 |
| | 4 | - | - | - | 23.75 | - | - | - | 53.49 | - | - | - | 49.64 | - | - | - | 51.21 |
| | | | $M = 1000$ | | | | $M = 5000$ | | | | $M = 15000$ | | | | $M = 50000$ | | |

Table 1: Tabulated values of $P_{nm}$, averaged over three identical runs, for the DQN baselines as a function of replay buffer size. Shown is the performance, measured on task $m < n$ after training on task $n$. For following performance evolution for a given task (rows in boxes) over the course of a given experiment (boxes), move along a row from left to right.

| benchmark $\rightarrow$ | Pushing-Objects | | Line-Following | | Robot Pursuit | |
|---|---|---|---|---|---|---|
| $\downarrow$ baseline | $P$ | $F$ | $P$ | $F$ | $P$ | $F$ |
| DQN+ER-1000 | 7.94 | 6.83 | 16.11 | 7.31 | 37.16 | 17.4 |
| DQN+ER-5000 | 10.84 | 3.14 | 17.62 | 5.61 | 47.88 | 5.37 |
| DQN+ER-15000 | 9.68 | 1.32 | **21.5** | -2.45 | 37.14 | 5.97 |
| DQN+ER-50000 | 11.62 | 0.12 | 20.76 | -2.14 | 49.58 | -5.03 |
| DQN+PER-1000 | 6.86 | 6.14 | 13.96 | 9.81 | 30.15 | 19.25 |
| DQN+PER-5000 | 10.07 | 3.02 | 19.27 | 2.48 | 50.91 | -6.38 |
| DQN+PER-15000 | 10.45 | 2.79 | 20.45 | 0.58 | 52.21 | -12.41 |
| DQN+PER-50000 | 10.73 | -0.5 | 21.55 | 0.6 | **53.84** | -2.26 |
| SFT | 4.85 | 12.3 | 5.71 | 15.2 | 15.43 | 34.86 |
| ARL | **14.17** | -2.91 | 19.13 | 3.21 | 53.54 | 0.55 |

Table 2: High-level performance measures for all benchmarks and baselines.

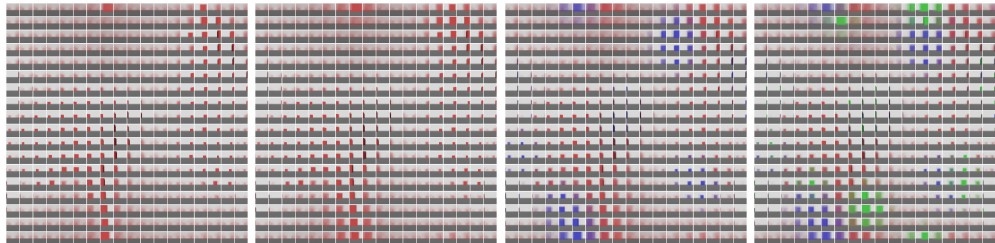

Figure 6: Left to right: GMM centroids visualized at the end of the babbling phase, task 1, task 2 and task 3. Task 4 is omitted since very few new centroids are learned. At each task, we observe the gradual embedding of new knowledge (blocks of new colors) into existing centroids. Best viewed in color and under magnification.

Higher precision and less forgetting are observed in table 2 for all benchmarks when using the largest buffer size, underscoring that really large buffers are required for combating forgetting. The fine-grained evaluation of table 1 then shows that, for the largest buffer size, this improvement is actually composed of two contributions in DDQN+ER experiments: less forgetting but also less learning of new tasks, to be observed in inferior $P_{33}$ and $P_{44}$ values for all benchmarks when comparing the largest to the smallest buffer sizes. Prioritized experience replay (PER) seems to remedy this problem and, indeed, improves results in general, but of course comes with its own set of tunable parameters on which performance critically depends.

The corresponding fine-grained ARL results are found in the first column of table 3. High-level ARL results show generally comparable or superior performance over the best DQN baselines with very large buffers, see table 2. The same table shows that SFT is not a feasible strategy for CRL at all, and the corresponding fine-grained performance values are not tabulated to save space. Generally, we observe that DNN-based baselines with large buffers show less retention but, in many cases, stronger backwards transfer expressed by negative forgetting measures. Backwards transfer, which implies that later task contribute to improvement in previous tasks, is a phenomenon rarely observed in supervised CL, seems to be a feature that is common in CRL. Since the results generally show standard deviations around 2.0, we may state that ARL can egalize the performance of the best baselines, however without resorting to replay buffers.

## 2.7 Qualitative Analysis of Representation Learning

In this set of experiments on ARL, we will visualize the samples with highest TD errors collected during a wake phase, as well as the representations arising from updating with these samples. This will be done for the pushing-objects benchmark only, since its samples (20x20x3 RGB images) have the easiest visual interpretation. An useful property of the GMMs employed for representation learning is that their component centroids "live" in the space of the data they model. This means they can themselves be interpreted and visualized as images, thus allowing to understand what has been learned. Indeed, this explainability property is an important advantage of GMMs over DNNs. In fig. 6, we show the centroids for a ratio of generated to real samples of 1 (easier for visualization since new samples have a stronger impact) and the usual AR parameters, at the end of the babbling phase as well as the end of tasks 1-4. For all tasks, we observe a gradual integration of new-task centroids into existing ones, in a way that shifts and redistributes existing centroids instead of simply replacing them. This is due to the selective replay mechanism, see section 2.3. In particular, we observe a shift towards larger blocks after processing task 1 w.r.t. the babbling phase, purely random motor babbling will, in general, not lead the robot close to a red block. Consequently, when large blocks are encountered, their associated TD error is high, leading to inclusion into the learning set for the sleep phase.

The samples that were used for obtaining these representations are shown in fig. 7, sorted by their associated TD error. We generally observe that observations close to a block have the highest TD-errors, since pushing a massive block, as well a pushing a massless block, give rise to $\pm 10$ reward and thus to large errors if this is not anticipated. We observe that task 4 blocks have a color similar

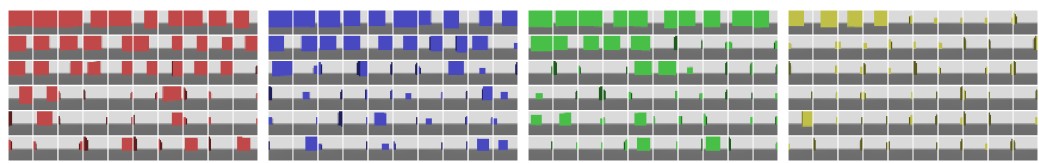

Figure 7: From left to right: the 60 highest TD-error samples collected during the first wake phase of PO tasks 1-4.

to task 3 blocks, so task 3 representations are presumably re-used, leading to a minimal learning of large task 4 blocks since not many of them have large TD errors.

## 2.8 ABLATION STUDY FOR ARL

In order to understand how the various ARL parameter contribute to the method's performance, we vary each relevant one individually while leaving the others constant. Obviously, we can do this only for the most relevant parameters, which we identified to be (see also section 2.3): the number of wake phases per task $T/C$, the percentage $\chi$ of highest-error transitions to be used for representation learning in sleep phases, the ratio of generated to real samples in ARL, and the initial adaptation radius $\sigma_0$ for the generator. Similarly to the preceding section, we present the detailed performance measure $P_{mn}$ for each benchmark and parameter setting. The observed picture is not very clear. Some parameter variations universally result in inferior performance, e.g., the number of sleep phases, for which the best value seems to be 1. Setting $\sigma_0$ too large also seems to be problematic, as is too high a ratio of generated to observed transitions, This parameter governs, among others, how important it is to preserve past knowledge w.r.t. acquiring new knowledge and is always hard to tune, even for generative replay models in supervised CL.

Give the observed variability of experimental outcomes, we may therefore conclude that most parameter variations have impact on performance although these, in general, do not result in a catastrophic loss of acquired policies.

## 2.9 DISCUSSION

**Choice of evaluation benchmarks** The chosen benchmarks were chosen because they include a potentially large number of tasks which have comparable difficulty, so the ordering of tasks does not impact the results. Furthermore, they allow expressive visualizations of the learned representations as shown in section 2.7. And lastly, their intrinsic difficulty is relatively low, so we can be sure that negative results are not caused by insufficient model complexity. Inspirations for these benchmarks were taken from other works on representation learning, see, e.g., Traoré et al. (2019). For establishing the basic capacity for continuous RL, these benchmarks therefore seem more appropriate to us than, e.g., Atari Games which, besides, contain a significant number of parameters on which results may critically depend (initialization for example).

**Choice of baselines** We concentrated on variants of Q-learning here because we wished to demonstrate a very basic effect, i.e, the strong reliance on large replay buffers with the associated affects on learning speed. Since more advanced RL variants like soft actor-critic (Haarnoja et al. (2018); Wolczyk et al. (2022a)) contain additional functions to be approximated by neural networks like the state-value function in SAC, the issue of catastrophic forgetting will be even more pronounced, require large buffers again, with the demonstrated consequences.

**CL methods for CRL: adiabatic -vs- experience replay** We experimentally verified that experience replay (ER) with large buffers is capable of dealing with the presented benchmarks. However, this implies a replay buffer large enough to store all transitions seen so far. It follows that the buffer size must scale linearly with the number of CRL tasks. While this might be acceptable when considering memory only, there is also time complexity to be considered: as shown in section 2.6, the learning of new tasks is slowed down by large buffers. Thus, the time until a new task has been properly acquired increases linearly with the number of previous tasks (i.e., buffer size), which might be considered unacceptable. In contrast, ARL just requires a single frozen model that is created on-

Table with evaluation results. Column groups are "eval. after task $m$" with sub-columns 1, 2, 3, 4 repeated for four parameter settings.

**Pushing-Objects (PO)**

| baseline | task $n$ | $\sigma_0=1$ 1 | 2 | 3 | 4 | $\sigma_0=0.5$ 1 | 2 | 3 | 4 | $\sigma_0=2$ 1 | 2 | 3 | 4 | $\sigma_0=3$ 1 | 2 | 3 | 4 |
|---|---|---|---|---|---|---|---|---|---|---|---|---|---|---|---|---|---|
| ARL with varying $\sigma_0$ | 1 | 6.63 | 9.79 | 6.79 | 9.9 | 3.46 | 4.5 | 0.29 | 7.27 | 9.55 | 9.5 | 10.04 | 9.87 | 0.98 | 3.3 | -2.78 | -2.13 |
| | 2 | - | 17.87 | 17.9 | 17.91 | - | 18.43 | 18.4 | 18.44 | - | 11.68 | 18.84 | 18.68 | - | 18.72 | 18.83 | 18.52 |
| | 3 | - | - | 1.53 | 10.08 | - | - | 9.45 | 4.46 | - | - | 7.0 | 8.94 | - | - | 8.88 | 9.01 |
| | 4 | - | - | - | 18.78 | - | - | - | 16.77 | - | - | - | 18.49 | - | - | - | 12.17 |

| baseline | task $n$ | $\chi=0.1$ 1 | 2 | 3 | 4 | $\chi=0.2$ 1 | 2 | 3 | 4 | $\chi=0.3$ 1 | 2 | 3 | 4 | $\chi=0.5$ 1 | 2 | 3 | 4 |
|---|---|---|---|---|---|---|---|---|---|---|---|---|---|---|---|---|---|
| ARL with varying $\chi$ | 1 | 6.63 | 9.79 | 6.79 | 9.9 | 3.41 | -0.87 | 2.54 | -2.14 | 10.12 | 8.96 | 10.2 | 10.53 | 10.91 | 9.87 | 9.82 | 10.09 |
| | 2 | - | 17.87 | 17.9 | 17.91 | - | 18.57 | 18.47 | 17.56 | - | 16.96 | 17.57 | 18.34 | - | 17.31 | 12.3 | 13.48 |
| | 3 | - | - | 1.53 | 10.08 | - | - | -2.7 | -1.57 | - | - | 9.68 | 9.46 | - | - | 6.32 | -1.09 |
| | 4 | - | - | - | 18.78 | - | - | - | 18.78 | - | - | - | 18.05 | - | - | - | 18.88 |

| baseline | task $n$ | 1 1 | 2 | 3 | 4 | 2 1 | 2 | 3 | 4 | 3 1 | 2 | 3 | 4 | – 1 | 2 | 3 | 4 |
|---|---|---|---|---|---|---|---|---|---|---|---|---|---|---|---|---|---|
| ARL with varying nr of sleep phases | 1 | 6.63 | 9.79 | 6.79 | 9.9 | 8.27 | 7.16 | 7.21 | -0.45 | 8.88 | 6.64 | 5.31 | -1.27 | - | - | - | - |
| | 2 | - | 17.87 | 17.9 | 17.91 | - | 16.98 | 18.26 | 12.24 | - | 16.96 | 17.57 | 15.3 | - | - | - | - |
| | 3 | - | - | 1.53 | 10.08 | - | - | 0.45 | 1.77 | - | - | 9.15 | 2.5 | - | - | - | - |
| | 4 | - | - | - | 18.78 | - | - | - | 18.35 | - | - | - | 18.99 | - | - | - | - |

| baseline | task $n$ | 2 1 | 2 | 3 | 4 | 1 1 | 2 | 3 | 4 | 3 1 | 2 | 3 | 4 | – 1 | 2 | 3 | 4 |
|---|---|---|---|---|---|---|---|---|---|---|---|---|---|---|---|---|---|
| ARL with varying ratio of real to generated transitions | 1 | 6.63 | 9.79 | 6.79 | 9.9 | 11.43 | 11.56 | 10.56 | 8.04 | 6.95 | 7.72 | 6.61 | 7.99 | - | - | - | - |
| | 2 | - | 17.87 | 17.9 | 17.91 | - | 17.25 | 17.34 | 16.51 | - | 12.64 | 15.38 | 14.61 | - | - | - | - |
| | 3 | - | - | 1.53 | 10.08 | - | - | 7.35 | 10.3 | - | - | 7.23 | 10.04 | - | - | - | - |
| | 4 | - | - | - | 18.78 | - | - | - | 18.57 | - | - | - | 17.55 | - | - | - | - |

**Line-Following (LF)**

| baseline | task $n$ | $\sigma_0=1$ 1 | 2 | 3 | 4 | $\sigma_0=0.5$ 1 | 2 | 3 | 4 | $\sigma_0=2$ 1 | 2 | 3 | 4 | $\sigma_0=3$ 1 | 2 | 3 | 4 |
|---|---|---|---|---|---|---|---|---|---|---|---|---|---|---|---|---|---|
| ARL with varying $\sigma_0$ | 1 | 22.08 | 21.61 | 20.4 | 15.49 | 23.51 | 22.14 | 12.23 | 10.34 | 22.81 | 21.65 | 22.38 | 19.61 | 23.45 | 17.81 | 19.43 | 22.95 |
| | 2 | - | 19.93 | 12.40 | 13.2 | - | 21.73 | 16.81 | 15.74 | - | 16.96 | 17.57 | 18.34 | - | 23.17 | 16.2 | 21.2 |
| | 3 | - | - | 19.43 | 20.24 | - | - | 22.32 | 7.46 | - | - | 19.15 | 22.5 | - | - | 21.2 | 19.99 |
| | 4 | - | - | - | 23.49 | - | - | - | 21.54 | - | - | - | 18.99 | - | - | - | 24.42 |

| baseline | task $n$ | $\chi=0.1$ 1 | 2 | 3 | 4 | $\chi=0.2$ 1 | 2 | 3 | 4 | $\chi=0.3$ 1 | 2 | 3 | 4 | $\chi=0.5$ 1 | 2 | 3 | 4 |
|---|---|---|---|---|---|---|---|---|---|---|---|---|---|---|---|---|---|
| ARL with varying $\chi$ | 1 | 22.08 | 21.61 | 20.4 | 15.49 | 23.51 | 22.14 | 12.23 | 10.34 | 22.81 | 21.65 | 22.38 | 19.61 | 23.45 | 17.81 | 19.43 | 22.95 |
| | 2 | - | 19.93 | 12.40 | 13.2 | - | 21.73 | 16.81 | 15.74 | - | 16.96 | 17.57 | 18.34 | - | 23.17 | 16.2 | 21.2 |
| | 3 | - | - | 19.43 | 20.24 | - | - | 22.32 | 7.46 | - | - | 19.15 | 22.5 | - | - | 21.2 | 19.99 |
| | 4 | - | - | - | 23.49 | - | - | - | 21.54 | - | - | - | 18.99 | - | - | - | 24.42 |

| baseline | task $n$ | 1 1 | 2 | 3 | 4 | 2 1 | 2 | 3 | 4 | 3 1 | 2 | 3 | 4 | – 1 | 2 | 3 | 4 |
|---|---|---|---|---|---|---|---|---|---|---|---|---|---|---|---|---|---|
| ARL with varying nr of sleep phases | 1 | 22.08 | 21.61 | 20.4 | 15.49 | 23.51 | 22.14 | 12.23 | 10.34 | 22.81 | 21.65 | 22.38 | 19.61 | - | - | - | - |
| | 2 | - | 19.93 | 12.40 | 13.2 | - | 21.73 | 16.81 | 15.74 | - | 16.96 | 17.57 | 18.34 | - | - | - | - |
| | 3 | - | - | 19.43 | 20.24 | - | - | 22.32 | 7.46 | - | - | 19.15 | 22.5 | - | - | - | - |
| | 4 | - | - | - | 23.49 | - | - | - | 21.54 | - | - | - | 18.99 | - | - | - | - |

| baseline | task $n$ | 2 1 | 2 | 3 | 4 | 1 1 | 2 | 3 | 4 | 3 1 | 2 | 3 | 4 | – 1 | 2 | 3 | 4 |
|---|---|---|---|---|---|---|---|---|---|---|---|---|---|---|---|---|---|
| ARL with varying ratio of real to generated transitions | 1 | 22.08 | 21.61 | 20.4 | 15.49 | 23.51 | 22.14 | 12.23 | 10.34 | 22.81 | 21.65 | 22.38 | 19.61 | - | - | - | - |
| | 2 | - | 19.93 | 12.40 | 13.2 | - | 21.73 | 16.81 | 15.74 | - | 16.96 | 17.57 | 18.34 | - | - | - | - |
| | 3 | - | - | 19.43 | 20.24 | - | - | 22.32 | 7.46 | - | - | 19.15 | 22.5 | - | - | - | - |
| 4 | | - | - | 23.49 | - | - | - | 21.54 | - | - | - | 18.99 | - | - | - | - | - |

**Robot Pursuit (RP)**

| baseline | task $n$ | $\sigma_0=1$ 1 | 2 | 3 | 4 | $\sigma_0=0.5$ 1 | 2 | 3 | 4 | $\sigma_0=2$ 1 | 2 | 3 | 4 | $\sigma_0=3$ 1 | 2 | 3 | 4 |
|---|---|---|---|---|---|---|---|---|---|---|---|---|---|---|---|---|---|
| ARL with varying $\sigma_0$ | 1 | 55.67 | 55.58 | 53.1 | 51.22 | 48.11 | 45.22 | 51.23 | 43.41 | 56.1 | 51.27 | 50.4 | 51.1 | 57.1 | 52.22 | 47.17 | 51.43 |
| | 2 | - | 48.12 | 49.10 | 52.45 | - | 55.67 | 52.44 | 48.32 | - | 49.57 | 52.34 | 51.1 | - | 53.17 | 49.22 | 52.72 |
| | 3 | - | - | 45.13 | 55.54 | - | - | 43.32 | 48.44 | - | - | 51.15 | 55.9 | - | - | 45.26 | 43.51 |
| | 4 | - | - | - | 56.1 | - | - | - | 52.05 | - | - | - | 56.12 | - | - | - | 55.42 |

| baseline | task $n$ | $\chi=0.1$ 1 | 2 | 3 | 4 | $\chi=0.2$ 1 | 2 | 3 | 4 | $\chi=0.3$ 1 | 2 | 3 | 4 | $\chi=0.5$ 1 | 2 | 3 | 4 |
|---|---|---|---|---|---|---|---|---|---|---|---|---|---|---|---|---|---|
| ARL with varying $\chi$ | 1 | 55.67 | 55.58 | 53.1 | 51.22 | 48.11 | 45.22 | 51.23 | 43.41 | 56.1 | 51.27 | 50.4 | 51.1 | 57.1 | 52.22 | 47.17 | 51.43 |
| | 2 | - | 48.12 | 49.10 | 52.45 | - | 55.67 | 52.44 | 48.32 | - | 43.21 | 49.57 | 52.34 | - | 53.17 | 49.22 | 52.72 |
| | 3 | - | - | 45.13 | 55.54 | - | - | 43.32 | 48.44 | - | - | 48.15 | 55.9 | - | - | 45.26 | 43.51 |
| | 4 | - | - | - | 56.1 | - | - | - | 52.05 | - | - | - | 56.12 | - | - | - | 55.42 |

| baseline | task $n$ | 1 1 | 2 | 3 | 4 | 2 1 | 2 | 3 | 4 | 3 1 | 2 | 3 | 4 | – 1 | 2 | 3 | 4 |
|---|---|---|---|---|---|---|---|---|---|---|---|---|---|---|---|---|---|
| ARL with varying nr of sleep phases | 1 | 55.67 | 55.58 | 53.1 | 51.22 | 48.11 | 45.22 | 51.23 | 43.41 | 56.1 | 51.27 | 50.4 | 51.1 | - | - | - | - |
| | 2 | - | 48.12 | 49.10 | 52.45 | - | 55.67 | 52.44 | 48.32 | - | 43.21 | 49.57 | 52.34 | - | - | - | - |
| | 3 | - | - | 45.13 | 55.54 | - | - | 43.32 | 48.44 | - | - | 49.15 | 55.9 | - | - | - | - |
| | 4 | - | - | - | 56.1 | - | - | - | 52.05 | - | - | - | 56.12 | - | - | - | - |

| baseline | task $n$ | 2 1 | 2 | 3 | 4 | 1 1 | 2 | 3 | 4 | 3 1 | 2 | 3 | 4 | – 1 | 2 | 3 | 4 |
|---|---|---|---|---|---|---|---|---|---|---|---|---|---|---|---|---|---|
| ARL with varying ratio of real to generated transitions | 1 | 55.67 | 55.58 | 53.1 | 51.22 | 48.11 | 45.22 | 51.23 | 43.41 | 56.1 | 51.27 | 50.4 | 51.1 | - | - | - | - |
| | 2 | - | 48.12 | 49.10 | 52.45 | - | 55.67 | 52.44 | 48.32 | - | 43.21 | 49.57 | 52.34 | - | - | - | - |
| | 3 | - | - | 45.13 | 55.54 | - | - | 43.32 | 48.44 | - | - | 51.15 | 55.9 | - | - | - | - |
| | 4 | - | - | - | 56.1 | - | - | - | 52.05 | - | - | - | 56.12 | - | - | - | - |

Table 3: ARL ablation study results. Tabulated values of $P_{nm}$ for the ARL as a function of the indicated parameters, see text. Shown is the performance, measured on task $m < n$ after training on task $n$. For following performance evolution for a given task (rows in boxes) over the course of a given experiment (boxes), move along a row from left to right.

the-fly in sleep phases, taking up a few hundred samples' worth of memory, plus a very small buffer storing high-TD transitions for the current wake phase. In addition, ARL always replays the same number of samples due to selective replay/updating, see section 2.3, so it has constant time complexity, independently of previous tasks.

## 3 CONCLUSIONS, GENERALITY OF RESULTS

A principal conclusion from this article is that RL can be conducted completely without replay buffers, and with high-TD-error samples only. This entails a fast reaction to changes in the environment which we model by CRL tasks, see section 2.1. CF is mitigated by using adiabatic replay, which, although requiring computational resources, can be performed at constant time complexity, a significant improvement w.r.t. existing CL approaches like generative replay. All of these advantages are obtained by replacing DNNs by adiabatic replay based on GMMs, i.e., by trading computational power of the employed models for the ability to learn continuously. Foundation models Verwimp et al. (2023) may be used to increase the computational power, or else deep hierarchical variants of AR based on Gepperth (2022). We believe that the ability to learn continuously is a very important ingredient when it comes to large-scale and long-term learning for truly human-level performance in future intelligent systems.

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
