# OpenReview forum: "Decoupled representation and policy acquisition for continual reinforcement learning"
_ICLR.cc/2025/Conference — ICLR 2025 Conference Withdrawn Submission_

### Official Review · Reviewer_V6oF · 2024-11-03

**Soundness:** 1
**Presentation:** 1
**Contribution:** 1
**Rating:** 1
**Confidence:** 3

**Summary:**

The authors introduces a method for continual reinforcement learning (CRL) named Adiabatic Reinforcement Learning (ARL). ARL aims to address the challenges of catastrophic forgetting (CF) in non-stationary environments by employing a wake-sleep approach that separates the learning of internal representations from policy acquisition. The method leverages adiabatic replay (AR), a selective internal replay mechanism, to mitigate forgetting without the need for large replay buffers.

**Strengths:**

No strengths. The paper is very hard to follow.

**Weaknesses:**

- The Introduction is too brief.
- Essential background information is presented without proper introduction. For instance, in line 61, it states, "see Krawczyk & Gepperth - -(2024b) for a discussion of why such balanced replay," and line 72 mentions, "The core of ARL is a replay-based CL method termed adiabatic replay (AR, see Krawczyk & Gepperth (2024a))." These points need to be elaborated.
- The Methods section disproportionately emphasizes the Benchmark; the authors should concentrate more on their techniques.
- Most citations are incorrectly formatted, confusing \citet and \citep.

**Questions:**

No questions.

---

### Official Review · Reviewer_mTnH · 2024-11-04

**Soundness:** 2
**Presentation:** 1
**Contribution:** 2
**Rating:** 3
**Confidence:** 3

**Summary:**

This paper introduces a method called Adiabatic RL (ARL) for tackling continual RL problems that can be decomposed into phasic tasks, with each task on its own (locally) being governed by stationary dynamics. The goal is to introduce a method that: (1) allows fast learning of incoming tasks, (2) avoids the need for large replay buffers to hold task-specific data, and (3) performs without direct identification of the new tasks within the observations (i.e. no encoding of task ID). At a high level, they enable this by adopting adiabatic replay (introduced by Krawczyk & Gepperth (2024a) in the context of continual supervised learning) and a Gaussian Mixture Model for representing the learner. They demonstrate the efficacy of this approach via experiments on a robotic simulator task based on a differential-drive model.

**Strengths:**

- I view the problem scenario of the paper of high importance: continual RL is an open and highly important research area, with problems such as loss of plasticity and catastrophic forgetting (among others) being actively explored by the community.

- I think the goals set by the authors are reasonable, and the approach has merits.

**Weaknesses:**

**Writing and paper's structure:**
Overall, paper reads like an early draft and needs to undergo revisions by the authors. I wouldn't have a problem with this if it didn't get in the way of readability and understanding of the core ideas and discussions. Some examples below:
- The structure and order of exposition feels a bit unnatural; e.g. Section 2 (Methods) starts with Benchmarks in Subsection 2.1, which itself contains several subsections around Line-Following, Pushing-Objects, Robotic Pursuit (~2 pages of content), then Subsection 2.2 discusses the Baselines, and finally Subsection 2.3 introduces the method. A better organization here would help first understand the core ideas of the paper prior to getting into the task scenarios, which since they are themselves not based on standard / known benchmarks, they consume time to understand on their own.
- Introduction doesn't provide a sufficient picture of the paper.
- Some minor errors are noted at the end of my review (these errors are minor but are throughout the paper.)

**Method exposition feels insufficient:**
I had to spend quite a long time to grasp what the method actually does, and still I'm not fully confident that my understanding is correct. I attempted to reach the codebase to ensure my understanding is correct (which in principle shouldn't be needed), but noticed that the link in the paper points to an empty repository. I believe adding an algorithm box is necessary. Also, an improved/more detailed discussion of the existing methods that are being combined into the proposed approach (especially the Adiabatic Replay method in its original context) would be of great benefit to the reader.

**Results are hard to follow:**
Using standard evaluation metrics, tasks, and mediums of presentation would greatly help this paper. Including a standard benchmark or illustrative task independent of the highly specialized differential drive problem setting would make it much easier to assess the impact of the propositions. For instance, tasks like those proposed in Wan et al. (2022) ("Towards Evaluating Adaptivity of Model-Based Reinforcement Learning Methods"; https://proceedings.mlr.press/v162/wan22d.html) could highly benefit the evaluations in this paper.

**Questions:**

1. Why is the proposed approach only evaluated in the specialized problem setting?

2. Wouldn't it be relevant to evaluate model-based RL approaches that are particularly devised to deal with similar CRL problem scenarios? (See, e.g., my earlier reference to Wan et al. (2022).)

3. Given that time limit of 25k is used to terminate the episodes (Sec. 2.1.1): are you using partial-episode bootstrapping to time-awareness (see Pardo et al. (2018) "Time limits in reinforcement learning")?


**Minor suggestions to improve readability:**
- L35: fig. 1 -> Fig. 1
- L40: "them **for to** the learner
- L90: \citep instead of \citet for Lesort et al. (2021)
- L92: \citep instead of \citet for Pfulb & Gepperth (2019)
- (citet is generally used in place of citep in many places)
- L135: footnote id should be placed after "simulator" without any space
- L135: Reference gz-transport is missing
- L137: Footnote should move to after period ".^2"
- L146: ".All friction": missing space
- (All references to figures should be using "Fig." in place of "fig.")
- 161: Footnote 2 is an empty repo
- 173-178: "line-following(LF, [...]": missing spaces (several instances in the same caption)
- L190: The actions space -> action

---

### Official Review · Reviewer_VX6w · 2024-11-05

**Soundness:** 3
**Presentation:** 2
**Contribution:** 2
**Rating:** 5
**Confidence:** 3

**Summary:**

This paper proposes adiabatic reinforcement learning to address catastrophic forgetting for continual learning, where high-error samples are leveraged instead of large replay buffers.

**Strengths:**

1. The motivation is very clear. The overall paper is easy to understand and comprehend
2. The innovative points of the paper are impressive, including fast learning of new tasks, low memory requirement and plastic representation.

**Weaknesses:**

1. The images describing the algorithm are difficult to understand, especially Fig. 4 and Fig. 5. Meanwhile, the proposed algorithm lacks pseudocode.
2. The proposed continuous learning mechanism has not been applied to more complex RL algorithms such as SAC or DSAC.
3. The testing environment is too simple and lacks comparison with the latest algorithms in continual RL. The authors are encouraged to apply multi-task learning of the same robot in DeepMind Control to enhance the convincibility of the experiment.

**Questions:**

1. For reinforcement learning tasks, is there any special improvement in the applied adiabatic replay method compared to that in (Krawczyk
& Gepperth (2024a))?
2. Are there any data or charts that can be displayed in the ablation test?
3. Compared with Prioritized Experience Replay in DQN, what are the significant differences in the high-error samples used in the paper?
4. There is too much content in the methods section. The content related to the experiment can be placed in a separate section to make the article structure more reasonable.

---

### Official Review · Reviewer_zK4f · 2024-11-06

**Soundness:** 2
**Presentation:** 1
**Contribution:** 2
**Rating:** 3
**Confidence:** 4

**Summary:**

The paper introduces a method that decouples learning the state representation and the policy (action-value function) for continual reinforcement learning. The proposed approach operates in two phases: the wake phase and the sleep phase. In the wake phase, the action-value function is quickly learned using samples that have high TD error, and during the sleep phase, the representations are slowly learned using adiabatic replay. The authors perform experiments on three robotics tasks where some benefits are observed for the proposed approach.

**Strengths:**

* The proposed approach is very promising and well-suited for continual reinforcement learning. It is intuitive to use a slow learner for representation learning and a fast learner for quick adaptation.
* The use of GMMs, a non-parametric model, for rapid learning is novel. I like the interpretability that results in its usage as demonstrated in Fig 6.

**Weaknesses:**

* The writing of the paper is unfortunately poor: many sentences are grammatically incorrect, there are several typos present throughout the paper, some notations are not defined, and the organization of the paper is unconventional (for e.g., the absence of the background section). Certain sections are hard to understand and misleading due to the below-par writing.
* The adiabatic replay technique, which is one of the major contributions of the paper, is not explained well. It is unclear what $p(s; \{\theta_k\})$ or $\pi_k$ is or how they are used. A good background section with more details on this technique is necessary to fully understand the proposed approach.
* The experiments are too simple to draw any meaningful conclusions. They provide a good validation (proof-of-concept) for the proposed approach, but they are very limited in order to draw broader conclusions. More experiments, using the standard CRL benchmarks such as gym-Minigrid, JellyBeanWorld, Procgen, or MetaWorld, should be performed to draw convincing conclusions.
* The results are quite hard to interpret or compare as they are presented as two long tables in hard-to-see font size, filled with lots of numbers.

**Questions:**

**Decision:**

The proposed idea is interesting and the initial results are promising. However, the paper is in its early stages and lacks proper writing and experimentation. Therefore, I recommend a **clear rejection** of the current draft.

**Areas of improvement:**

The idea of fast and slow learning is explored in many ways in RL and continual RL [e.g., 1, 2, and 3]. These concepts are inspired by complementary learning systems theory from neuroscience [4]. I strongly recommend the authors frame their approach under this lens and make connections with broad research that already exists. I also believe the PT-DQN that was proposed in [3] is a strong baseline to compare against as these two approaches are very similar, where some components are learnt fast and others are learnt slow.

**Questions:**

* Does your approach require the task boundary or the task ID information? How can one adapt the method when the non-stationarities are continuous?
* In Sec 2.3, what are $p(s; \{\theta_k\})$ and $\pi_k$? How is it used in the adiabatic replay?
* In Fig 6, the centroids learned for new colours seem to overlap significantly with the existing ones. Why do we observe this behaviour as opposed to learning a different centroid for various colours as colour is the only changing variable between tasks?

**References:**

[1] Botvinick, Matthew, et al. "Reinforcement learning, fast and slow." Trends in cognitive sciences 23.5 (2019): 408-422.

[2] Anand, Nishanth, and Doina Precup. "Prediction and control in continual reinforcement learning." Advances in Neural Information Processing Systems 36 (2024).

[3] Chung, Wesley. "Two-timescale networks for nonlinear value function approximation." (2019).

[4] Kumaran, Dharshan, Demis Hassabis, and James L. McClelland. "What learning systems do intelligent agents need? Complementary learning systems theory updated." Trends in cognitive sciences 20.7 (2016): 512-534.

---

### Note · Authors · 2024-11-14

I have read and agree with the venue's withdrawal policy on behalf of myself and my co-authors.